# New dual method for elastica regularization

**Jintao Song**[1], **Huizhu Pan**[2], **Jieyu Ding**[1], **Weibo Wei**[1], **Zhenkuan Pan**[1]*

**1** College of Computer Science and Technology, Qingdao University, Qingdao, Shandong, China, **2** School of Electrical Engineering, Computing and Mathematical Sciences, Curtin University, Perth, WA, Australia

* zkpan@126.com

**Data Availability Statement:** All relevant data are within the manuscript.

**Funding:** The author thanks the support of National Natural Science Foundation of Shandong Province (No.ZR2019LZH002).

## Abstract

The Euler's elastica energy regularizer has been widely used in image processing and computer vision tasks. However, finding a fast and simple solver for the term remains challenging. In this paper, we propose a new dual method to simplify the solution. Classical fast solutions transform the complex optimization problem into simpler subproblems, but introduce many parameters and split operators in the process. Hence, we propose a new dual algorithm to maintain the constraint exactly, while using only one dual parameter to transform the problem into its alternate optimization form. The proposed dual method can be easily applied to level-set-based segmentation models that contain the Euler's elastic term. Lastly, we demonstrate the performance of the proposed method on both synthetic and real images in tasks image processing tasks, i.e. denoising, inpainting, and segmentation, as well as compare to the Augmented Lagrangian method (ALM) on the aforementioned tasks.

## 1 Introduction

Traditional variational methods have been extensively applied to image restoration problems based on image features, such as texture, edge, and region, etc. [1–4]. In particular, the combination of the high-order TV term and Euler regularizers in variational models addresses certain problems that cannot be addressed by low-order models. However, the complexity of the terms makes the models more difficult to implement. It has become a challenge in recent years to design simpler and more effective solutions for the combined model. Before presenting our new work, we introduce the TV term and the Euler's elastica term below.

The problem of image restoration is finding the restored image $u = u(x)$ given the damaged image $f = f(x)$, $x \in \Re^d$, where $\Re$ is a bounded domain with a Lipschitz boundary. For a grayscale image, the image repair model is $f = Au + \eta$, where $\eta$ is the noise information and $A$ is a blurring operator [5–7]. The image restoration problem can then be formulated as the minimization of the following energy functional,

$$\min_u E(u) = \int_\Omega (Au - z)^2 dx. \tag{1}$$

However, the minimization problem in (1) is ill-posed. To solve this, Tikhonov et. al [8] proposed a regularization technique. By adding a smoothing regularizer into the energy functional, the problem will obtain a unique solution. The side effect is that the model can no

**Competing interests:** The authors have declared that no competing interests exist.

longer preserve edges in the image. The later proposed Rudin-Osher-Fatemi (ROF) model retains image edges by solving for a piecewise constant function in the space of bounded variation functions (BV). Nowadays, many methods based on TV regularization are used to deal with imaging problems such as image denoising [9–12] and image segmentation [13, 14].

Another downside of the TV models is that results are often accompanied by blocky (staircase) effects and loss of image contrast [15–18]. Recently, scholars have proposed many solutions such as iterative regularization techniques [19] and the use of other high-order terms to mitigate the problem. Faster implementations have also been invented [20].

As for the Euler's elastica [21], it has attracted much attention due to its good properties in mathematical and physical systems. The Euler's elastica energy functional is defined as

$$TeeV\,(u) = \int_{\Omega} \left(a + (\nabla \cdot n)^2\right) |\nabla u| dx, \tag{2}$$

where, $\boldsymbol{n} = \frac{\nabla n}{|\nabla n|}$. The term was first used in computer vision by Mumford [22], and has since proven to be effective in solving the problems present in the TV model. The Euler's elastica has also been widely applied in various fields of image processing such as image denoising [23–25], image segmentation [20, 26–28], inpainting [24, 29–31], illusory contour [32, 33], and segmentation with depth [34–36]. Therefore, we believe it is important to design an efficient numerical solution for the combined model.

Due to the non-convexity, non-smoothness, and high-order of the derivatives of 2, it is a challenging task to design a fast and efficient solution. The ALM method has achieved good results in optimizing 2, Tai et al. [23] first proposed this ALM method to solve the image inpainting problems, then Zhu et al. [28] extend the ALM method to image segmentation field. So far, the primal-dual technique [20, 37] performs better in optimizing 2.

In this paper, we propose a new primal-dual method for the solution of 2, that makes it easier to use. The key points of the proposed method can be summarised below: (i) We introduce the dual variable $\boldsymbol{p}$ to circumvent the curvature term. (ii) Using appropriate indicator functionals, we reformulate 2 as a minimization problem of $u$ and a maximization of $\boldsymbol{p}$. (iii) The subproblem $u$ has an analytic solution, and the subproblem $\boldsymbol{p}$ can be solved by a gradient descent algorithm. Numerical experiments demonstrate the improved efficiency of the proposed method.

Compared to the ALM method, the advantages of this method can be summarized into three points, (i) The proposed method only introduces one dual variable $\boldsymbol{p}$, while the ALM method introduces 8 intermediate variables. (ii) Due to the fewer variables, the proposed method has a weaker dependency on parameters. (iii) The CPU running time required for each iteration is greatly reduced.

The rest of this article is organized as follows. In the next section, we introduce the previous models and the associated numerical algorithms. In section 4, we propose our model for image denoising and image segmentation. The subproblems of energy minimization are solved in section 5. In section 6, we provide some numerical results to illustrate the effectiveness of the new algorithm. The last section presents the conclusions.

## 2 The previous works

### 2.1 The TV model for image denoise and inpainting

The well-known TV model [2] for image denoising is an energy minimization problem on $u$, such that

$$\min E(u) = \frac{1}{2} \int_{\Omega} (f - u)^2 dx + \gamma \int_{\Omega} |\nabla u| dx, \tag{3}$$

where, $\gamma$ is a penalty parameter for the summed length of the curves. To use the model in image inpainting, we need to incorporate a mask function $\eta$, which is defined as

$$\eta(x) = \begin{cases} 0, & \text{if } x \in x_L \\ 1, & \text{otherwise} \end{cases}, \tag{4}$$

where, $x_L = \{x_1, x_2 \ldots x_l\}$ denote the damaged regions. The classical TV model combined with this mask function is

$$\min E(u) = \frac{\eta}{2} \int_\Omega (f - u)^2 dx + \gamma \int_\Omega |\nabla u| dx. \tag{5}$$

It is evident that if $\eta$ is the identity matrix, then the inpainting model above is the same as the denoising model. Therefore, we only focus on the image restoration model. The evolution equation of $u$ can be derived via variational methods as

$$\begin{cases} \dfrac{\partial u(x, t)}{\partial t} = \eta(f - u) - \gamma \left( \nabla \cdot \dfrac{\nabla u}{|\nabla u|} \right) & t > 0, \ x \in \Omega \\[2mm] \dfrac{\partial u(x, t)}{\partial N} = 0 & t > 0, \ x \in \partial\Omega \\[2mm] u(x, 0) = u^0(x) & t = 0, \ x \in \Omega \end{cases}. \tag{6}$$

## 2.2 The TV model reformulated via the dual method

To simplify the TV model (3), we introduce the dual variable $p$ to circumvent the curvature term. Substituting $p$ into the TV model, we get

$$\min_u \max_{|p| \leqq \gamma} E(u, p) = \frac{1}{2} \int_\Omega (f - u)^2 dx + \int_\Omega (\nabla \cdot p) \, u \, dx. \tag{7}$$

By using the dual method, we can successfully avoid the curvature term and significantly simplify calculations. The new evolution equations of $u$ are

$$\begin{cases} \dfrac{\partial u(x, t)}{\partial t} = f - u + \nabla \cdot p & t > 0, \ x \in \Omega \\[2mm] \dfrac{\partial u(x, t)}{\partial N} = 0 & t > 0, \ x \in \partial\Omega \\[2mm] u(x, 0) = u^0(x) & t = 0, \ x \in \Omega \end{cases}, \tag{8}$$

and $p$ can be solved from

$$\begin{cases} \dfrac{\partial p(x, t)}{\partial t} = \nabla u \\[2mm] |p| \leqq \gamma \end{cases}. \tag{9}$$

## 2.3 The Euler's elastica model for image denoising

In order to recover edges and counter the staircase effect, Tai et al. [23] proposed the Euler's elastica model

$$\min E(u) = \frac{1}{2} \int_\Omega (f - u)^2 dx \int_\Omega \left( a + (\nabla \cdot n)^2 \right) |\nabla u| dx, \tag{10}$$

$$s.t. \ w = \nabla u, \ |w| = w \cdot m, \ m = n, \ q = \nabla \cdot n, \ |m| \leq 1.$$

The boundary produced by this method is curved rather than straight. Its solution via the ALM proposed by Tai et al. [23] simplifies the calculation and increases the optimization efficiency. The Tai-Hahn-Chung (THC) formulation is

$$
\begin{aligned}
E(u, \boldsymbol{w}, \boldsymbol{n}, \boldsymbol{m}, q, p) = {}& \\
\frac{1}{2}\int_\Omega (f-v)^2 dx + \int_\Omega & \left(a + (\nabla \cdot \boldsymbol{n})^2\right)|\nabla u| dx \\
+ \int_\Omega \lambda_1(|\boldsymbol{w}| - \boldsymbol{w}\cdot \boldsymbol{m})\, dx + {}& \gamma_1 \int_\Omega (|\boldsymbol{w}| - \boldsymbol{w}\cdot \boldsymbol{m})\, dx \\
+ \int_\Omega \boldsymbol{\lambda}_2 \cdot (\boldsymbol{w} - \nabla u)\, dx + {}& \frac{\gamma_2}{2}\int_\Omega |\boldsymbol{w} - \nabla u|^2 dx \\
+ \int_\Omega \lambda_3(v - u)\, dx + {}& \gamma_3 \int_\Omega (v-u)^2 dx \\
+ \int_\Omega \boldsymbol{\lambda}_4 \cdot (\boldsymbol{n} - \boldsymbol{m})\, dx + {}& \frac{\gamma_4}{2}\int_\Omega |\boldsymbol{n} - \boldsymbol{m}|^2 dx + \delta_{\mathcal{R}}(\boldsymbol{m})
\end{aligned}
\tag{11}
$$

where, the functions $\delta_{\mathcal{R}}(\boldsymbol{v})$ and $\delta_{\mathcal{R}}(\boldsymbol{m})$ denote the constraints $|\boldsymbol{m}| \leq 1$ and $0 \leq u \leq 1$ respectively.

## 2.4 The Chan-Vese model with elastica for image segmentation

The task of two-phase segmentation $f(x): \Omega \to R$ of a gray value image is to divide the image into two regions $\Omega_1, \Omega_2$. The Chan-Vese model, a classical two-phase segmentation model, [38] is a reduced piecewise constant Mumford-Shah model [4] under the variational level set framework. Its form is

$$
\begin{aligned}
\min E(c_1,\ c_2,\ \phi) = {}& \int_\Omega (f - c_1)^2 H_\varepsilon(\phi)\, dx \\
+ \int_\Omega (f - c_2)^2 (1 - H_\varepsilon(\phi))\, dx + {}& \gamma \int_\Omega |\nabla H_\varepsilon(\phi)| dx, \\
s.t.\ |\nabla \phi| = 1. &
\end{aligned}
\tag{12}
$$

In the above model, $\phi$ is the level set function and $H(\phi)$ is the Heaviside function of $\phi(x)$, stated as

$$
H(\phi(x)) = \begin{cases} 1, & \text{if } \phi(x) \geq 0 \\ 0, & \text{otherwise} \end{cases}.
\tag{13}
$$

Replacing the TV regularizer with the Eular's elastica energy in the Zhu-Tai-Chan (ZTC) formulation [28] leads to the Chan-Vese model with elastica (CVE) below

$$
\begin{aligned}
\min E(c_1,\ c_2,\ \phi) = {}& \int_\Omega Q(c_1,\ c_2) H_\varepsilon(\phi) dx \\
+ \gamma \int_\Omega \left( a + b\left( \nabla \cdot \frac{\nabla H(\phi)}{|\nabla H(\phi)|} \right)^2 \right) & |\nabla H_\varepsilon(\phi)| dx, \\
s.t.\ |\nabla \phi| = 1, &
\end{aligned}
\tag{14}
$$

where, $Q(c_1, c_2) = \alpha_1(c_1 - f)^2 - \alpha_2(c_2 - f)^2$. Adding another variable that relaxes the Heaviside function, $u = H(x)$ and $u \in [1, 0]$, we can construct the following augmented Lagrangian functional

$$E(u, \boldsymbol{w}, \boldsymbol{n}, \boldsymbol{m}, q, p) =$$

$$\int_\Omega Q(c_1, \ c_2) \ vdx + \gamma \int_\Omega [a + b|\nabla \cdot \boldsymbol{n}|^2] |\boldsymbol{w}| dx$$

$$+ \int_\Omega \lambda_1 (|\boldsymbol{w}| - \boldsymbol{w} \cdot \boldsymbol{m}) \ dx + \gamma_1 \int_\Omega (|\boldsymbol{w}| - \boldsymbol{w} \cdot \boldsymbol{m}) \ dx$$

$$+ \int_\Omega \lambda_2 \cdot (\boldsymbol{w} - \nabla u) \ dx + \frac{\gamma_2}{2} \int_\Omega |\boldsymbol{w} - \nabla u|^2 dx \qquad (15)$$

$$+ \int_\Omega \lambda_3 (v - u) \ dx + \gamma_3 \int_\Omega (v - u)^2 dx + \delta_{\mathcal{R}}(v)$$

$$+ \int_\Omega \lambda_4 \cdot (\boldsymbol{n} - \boldsymbol{m}) \ dx + \frac{\gamma_4}{2} \int_\Omega |\boldsymbol{n} - \boldsymbol{m}|^2 dx + \delta_{\mathcal{R}}(\boldsymbol{m})$$

The functions $\delta_{\mathcal{R}}(v)$ and $\delta_{\mathcal{R}}(\boldsymbol{m})$ denote the constraints $|\boldsymbol{m}| \le 1$ and $0 \le u \le 1$ respectively.

## 3 The CVE model reformulated via the dual method

### 3.1 Image denoise and inpainting

Combining (7) and (10), we propose the dual formulation of the Chan-Vese model with elastica

$$\min_u \max_{|p| \leqq g} E(u, p) = \frac{\eta}{2} \int_\Omega (f - u)^2 dx + \gamma \int_\Omega (\nabla \cdot \boldsymbol{p}) \ u \ dx,$$

$$s.t. \ g = \left( a + b \left( \nabla \cdot \frac{\nabla u}{|\nabla u|} \right)^2 \right) \qquad (16)$$

This minimization problem can be divided into two subproblems, and their solutions can be expressed as follows:

**$u - subproblem$**. This subproblem is a minimization problem, and the objective function of optimization is

$$\frac{\eta}{2} \int_\Omega (f - u)^2 dx + \gamma \int_\Omega (\nabla \cdot \boldsymbol{p}) \ u \ dx, \qquad (17)$$

We find that this function is almost identical to the TV model and get the same solution:

$$\begin{cases} \dfrac{\partial u(x, \ t)}{\partial t} = \eta(f - u) + \gamma \nabla \cdot \boldsymbol{p} & t > 0, \ x \in \Omega \\[2mm] \dfrac{\partial u(x, \ t)}{\partial N} = 0 & t > 0, \ x \in \partial\Omega \\[2mm] u(x, \ 0) = u^0(x) & t = 0, \ x \in \Omega \end{cases} \qquad (18)$$

In this formula, $u$ has an analytic solution, so there is no need to iterate further and running time is greatly reduced.

***p − subproblem***. The dual variable **p** can be solved by

$$\begin{cases} \dfrac{\partial \boldsymbol{p}(x,\ t)}{\partial t} = \gamma \nabla u \\[2mm] |\boldsymbol{p}| \leqq g \end{cases}. \tag{19}$$

where the value of $g$ is directly determined by the $u$ obtained in the previous step.

This is a good way to avoid fourth-order terms and can solve the Euler's elastica term better. Since no additional parameters are introduced, and the iteration time required for each step is greatly reduced compared to the ALM solution.

## 3.2 Image segmentation

Combining (7) and (14), we propose the Chan-Vese model with elastica reformulated with the dual method shown below,

$$\min_{u} \max_{|p| \leqq g} E(u, p) = \int_{\Omega} Q(c_1,\ c_2)\, u dx + \gamma \int_{\Omega} (\nabla \cdot \boldsymbol{p})\, u\ dx.$$

$$s.t.\ g = \left( a + b \left( \nabla \cdot \frac{\nabla u}{|\nabla u|} \right)^{2} \right) \tag{20}$$

***$c_1$, $c_2$ − mean value***. The Chan-Vese model is a two-term segmentation model, $c_1$ and $c_2$ are mean values of the foreground and background,

$$c_1^{k+1} = \frac{\int_{\Omega} f(x) H(\phi^k(x))\, dx}{\int_{\Omega} H(\phi^k(x))\, dx},$$

$$c_2^{k+1} = \frac{\int_{\Omega} f(x)(1 - H(\phi^k(x)))\, dx}{\int_{\Omega} (1 - H(\phi^k(x)))\, dx}. \tag{21}$$

***u − subproblem***. Similar to (18), we can also obtain an exact solution for $u$,

$$\begin{cases} \dfrac{\partial u(x,\ t)}{\partial t} = Q + \gamma \nabla \cdot \boldsymbol{p} & t > 0,\ x \in \Omega \\[2mm] \dfrac{\partial u(x,\ t)}{\partial N} = 0 & t > 0,\ x \in \partial\Omega \\[2mm] u(x,\ 0) = u^0(x) & t = 0,\ x \in \Omega \end{cases}. \tag{22}$$

***p − subproblem***. Although the meaning of u has changed, p can still be solved the same way as (19).

In this section, we presented the dual formulation of the CVE model for image denoising and segmentation. Next, we will design the discretized numerical algorithms for the models.

## 4 Numerical implementations of the sub-problems of minimization

### 4.1 Image denoising and inpainting

To compute the two subproblems numerically, we need to design discrete algorithms for each problem. For the sake of simplicity, we discretize the image domain pixel by pixel with the rows and column numbers as indices. Then, the gradients can be represented approximately

by forward, backward, and central finite differences,

$$\nabla^+ u_{i,j} = \begin{bmatrix} \partial^+_{x_1} u_{i,j} \\ \partial^+_{x_2} u_{i,j} \end{bmatrix}, \quad \nabla^- u_{i,j} = \begin{bmatrix} \partial^-_{x_1} u_{i,j} \\ \partial^-_{x_2} u_{i,j} \end{bmatrix},$$

$$\nabla^o u_{i,j} = \begin{bmatrix} \partial^o_{x_1} u_{i,j} \\ \partial^o_{x_2} u_{i,j} \end{bmatrix},$$

where,

$$\begin{cases} \partial^+_{x_1} u_{i,j} = u_{i+1,j} - u_{i,j} \\ \partial^-_{x_1} u_{i,j} = u_{i,j} - u_{i-1,j} \end{cases}, \quad \begin{cases} \partial^+_{x_2} u_{i,j} = u_{i,j+1} - u_{i,j} \\ \partial^-_{x_2} u_{i,j} = u_{i,j} - u_{i,j-1} \end{cases}, \quad \begin{cases} \partial^o_{x_1} u_{i,j} = \frac{1}{2}\left(u_{i+1,j} - u_{i-1,j}\right) \\ \partial^o_{x_2} u_{i,j} = \frac{1}{2}\left(u_{i,j+1} - u_{i,j-1}\right) \end{cases}.$$

The Euler's Elastica term of $u$ can be stated as

$$\begin{cases} \nabla^- \cdot \dfrac{\nabla^+ u_{i,j}}{|\nabla^+ u_{i,j}|} = \nabla^- \cdot \left(\boldsymbol{w}_{i,j}\right) \\ \\ with \quad \boldsymbol{w} = \dfrac{u_{i+1,j} + u_{i,j+1} - 2u_{i,j}}{|u_{i+1,j} + u_{i,j+1} - 2u_{i,j}|} \end{cases}. \tag{23}$$

The other variables can be expressed in similar ways. Next, we will give a detailed explanation of the solutions to the subproblems obtained in section 4.

$u - subproblem$. The partial derivative with respect to $E$ gives the analytic solution with respect to $u$ as follows

$$u_{i,j}^{k+1,l+1} = f_{i,j}^{k+1} + \gamma \nabla^0 \cdot \boldsymbol{p}_{i,j}^k. \tag{24}$$

The $u$ in the image inpainting model can be formulated as follows,

$$u_{i,j}^{k+1,l+1} = f_{i,j}^{k+1} + \frac{\gamma \nabla^0 \cdot \boldsymbol{p}_{i,j}^k}{\eta}. \tag{25}$$

In fact, this formulation is the same as the corresponding one in the solution of the TV model using the dual method.

$p - subproblem$. The dual variable $\boldsymbol{p}$ also can be solved by

$$\begin{cases} \boldsymbol{p}_{i,j}^{k+1} = \boldsymbol{p}_{i,j}^k + t\nabla^0 u \\ \\ \boldsymbol{p}_{i,j}^{k+1} = \dfrac{\left(a + b\left(\nabla^- \cdot \dfrac{\nabla^+ u}{|\nabla^+ u|}\right)^2\right) \cdot \boldsymbol{p}}{\max\left\{\left(a + b\left(\nabla^- \cdot \dfrac{\nabla^+ u}{|\nabla^+ u|}\right)^2\right), |\boldsymbol{p}|\right\}} \end{cases}, \tag{26}$$

where no additional parameters have been introduced compared to the ALM, so the iteration time required for each step is greatly reduced.

In each iteration, the following error tolerance should be checked to determine convergence, i. e.,

$$\Sigma^{k+1} \leq \text{Tol}, \tag{27}$$

where, Tol = 0.001. $\Sigma^{k+1}$ are defined as

$$\Sigma^{k+1} = \frac{\|E^{k+1} - E^k\|}{\|E^k\|}. \tag{28}$$

In summary, the denoising al-gorithm is shown in Algorithm 1, and the inpainting algorithm is shown in Algorithm 2.

**Algorithm 1**: Dual elastica for denoising

```
1: Initialization: Set a = 3, b = 1, γ = 1, t = 0.0125.
2: while any stopping criterion is not satisfied do
   Calculate u^{k+1} from (24)
   Calculate p^{k+1} from (26)
3: end while
```

**Algorithm 2**: Dual elastica for inpainting

```
1: Initialization: Set a = 3, b = 1, γ = 10, t = 0.0125.
2: while any stopping criterion is not satisfied do
   Calculate u^{k+1} from (25)
   Calculate p^{k+1} from (26)
3: end while
```

## 4.2 Image segmentation

In this subsection, we will apply discretization to solve the formulas obtained in the previous image segmentation section.

$c_1, c_2$ – **mean value**. (21) can be solved by

$$c_1^{k+1} = \frac{\sum_{i=1}^{M}\sum_{j=1}^{N} f_{i,j} u_{i,j}^k}{\sum_{i=1}^{M}\sum_{j=1}^{N} u_{i,j}^k}, c_2^{k+1} = \frac{\sum_{i=1}^{M}\sum_{j=1}^{N} f_{i,j}(1 - u_{i,j}^k)}{\sum_{i=1}^{M}\sum_{j=1}^{N}(1 - u_{i,j}^k)}. \tag{29}$$

$u$ – **subproblem**. Same as TV model, $u$ in (22) can be computed by

$$u_{i,j}^{k+1,l+1} = Q_{i,j}^{k+1} + \gamma\nabla^0 \cdot \boldsymbol{p}_{i,j}^k. \tag{30}$$

$p$ – **subproblem**. Here, the Euler's elastic term is solved directly by the differential equation. We can get

$$\begin{cases} \boldsymbol{p}_{i,j}^{k+1} = \boldsymbol{p}_{i,j}^k + t\nabla \cdot \boldsymbol{w}_{i,j}^{k+1} \\ \\ \boldsymbol{p}_{i,j}^{k+1} = \dfrac{[a + b|\nabla^- \cdot \dfrac{\nabla^+ u}{|\nabla^+ u|}|^2] \cdot \boldsymbol{p}_{i,j}}{\max\{[a + b|\nabla^- \cdot \dfrac{\nabla^+ u}{|\nabla^+ u|}|^2], |\boldsymbol{p}_{i,j}|\}} \end{cases}. \tag{31}$$

Algorithm 3 is the summary of the dual elastica segmentation method.

**Algorithm 3**: Dual elastica for segmentation

```
1: Initialization: Set a = 0.001, b = 5, γ = 2, t = 0.0125.
2: while any stopping criterion is not satisfied do
   Calculate u^{k+1} from (30)
   Calculate p^{k+1} from (31)
3: end while
```

## 5 Numerical experiments

In this section, we show the results from numerical simulations to illustrate the effectiveness of our algorithms in image denoising, inpainting, and segmentation. All experiments are running in MATLAB R2020a.

The parameters in (11) were set by $a = 1$, $b = 1$, $\gamma = 100$, $\gamma_1 = 7$, $\gamma_2 = 20$, $\gamma_3 = 5$, $\gamma_4 = 15$ for all denoising experiments. In (15), the parameters were set to $\mu_1 = 0.6$, $\mu_2 = 1$, $a = 0.1$, $b = 2$, $\gamma = 5$, $\gamma_1 = 1.4$, $\gamma_2 = 10$, $\gamma_3 = 5$, $\gamma_4 = 5$ for all segmentation experiments.

### 5.1 Testing of Algorithm 1 and Algorithm 2

First of all, we tested our algorithm on three synthetic images. The size of the synthetic images is 256 x 256.

The image denoising result is shown in Fig 1, where (a) is the picture of a clipped triangle with Gaussian noise, (b) is the result of Algorithm 1, and (c) is the relative error, i.e. noise removed through denoising. Fig 2 shows the results of the inpainting experiment, where Fig 2 (a) shows a circle corrupted by white areas, (b) to (f) are the results at iterations $k = 100, 200, 300, 400$, and 500 using Algorithm 2. It is evident that the results improved over time. Fig 3 shows the results of segmentation, where (a) is the initialization and (b) is the result of Algorithm 3, (c) is the clean segmentation contour, and (d) shows the level set function $\phi$.

Fig 4 shows the results of image denoising and inpainting by the TV method and the proposed dual elastica algorithm. Column (a) shows the original noisy images, (b) and (c) are the denoising results of the TV method and the dual Euler's elastic algorithm, respectively. By observing the details, we can see that the our proposed algorithm can remove noises effectively without staircase effect and performed better in preserving boundaries compared to the TV model.

Fig 5 shows the main difference between the TV model and the Euler's elastica model which is that the Euler's elastica model performed much better in preserving boundary and corners.

Besides the qualitative performance of our algorithm in image deniosing, inpainting, and segmentation, computational efficiency is another major point of interest. In Fig 1, the algorithm took 9 iterations to satisfy (28) and the CPU running time was 0.0230 seconds to achieve a PSNR score of 27.9992. In Fig 2, our algorithm ran for 500 iterations and each iteration took 0.0034 seconds, so the total CPU time was 1.7048 seconds. The last segmentation experiment

(a)                          (b)                          (c)

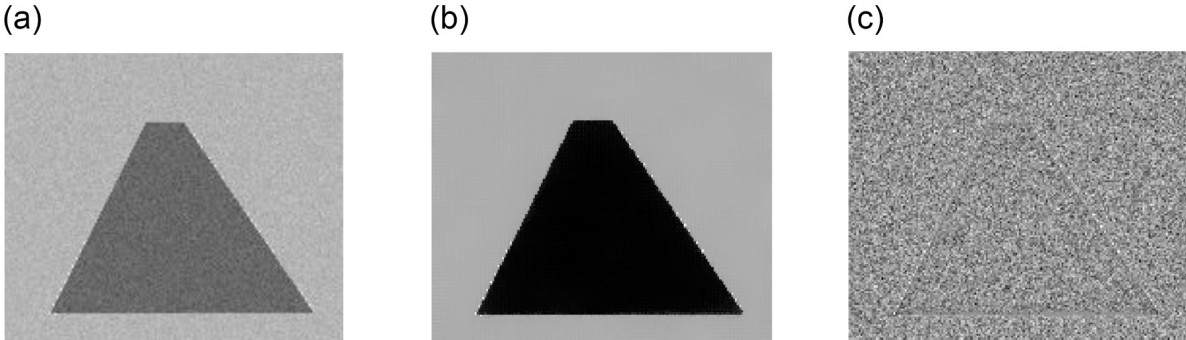

**Fig 1. The denoising results for synthetic images.** (a) is the original noisy picture, (b) is the result of the proposed method, (c) is the relative error.

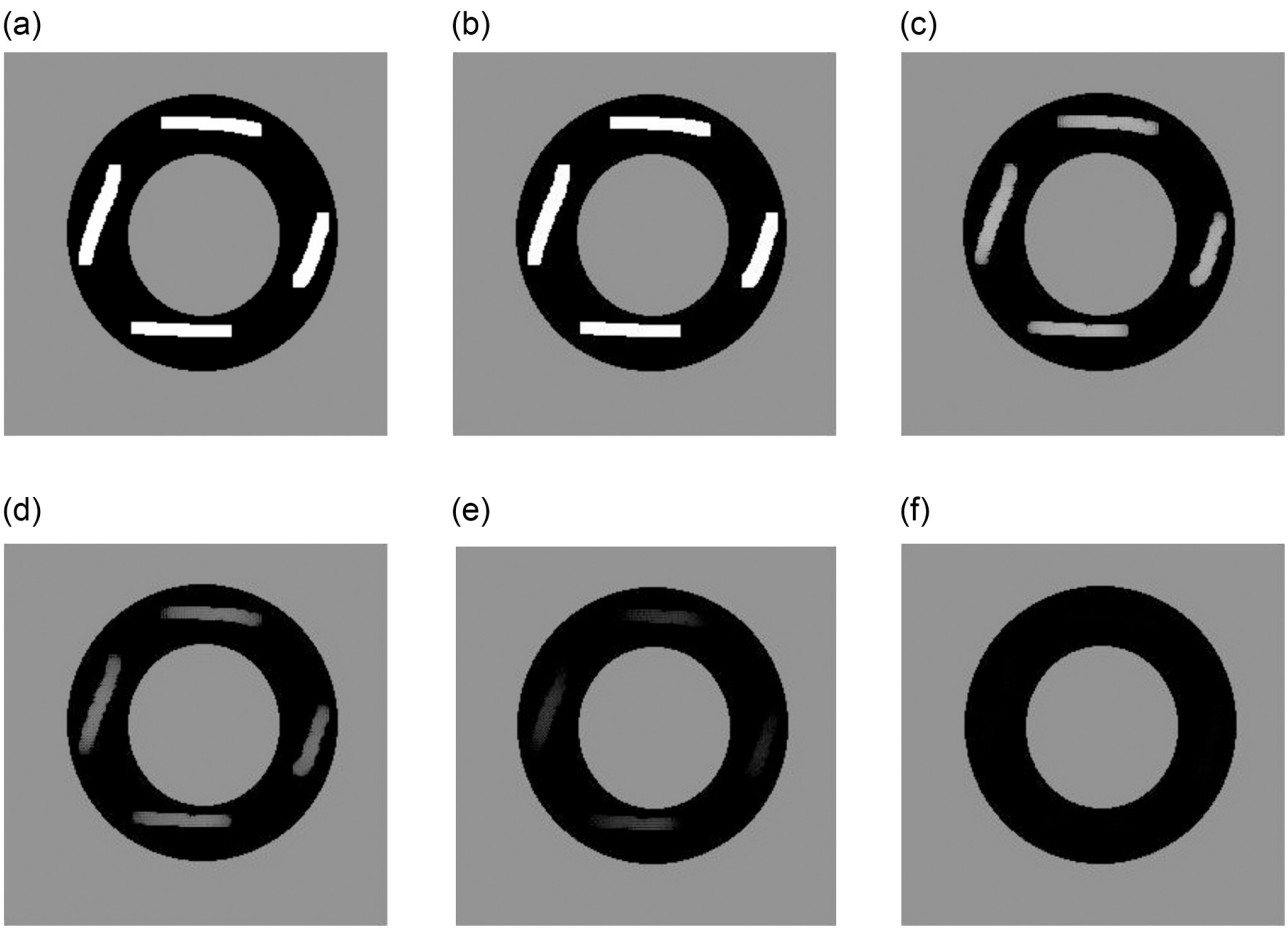

**Fig 2. The inpainting results for synthetic images.** (a) is the original broken picture, (b) to (f) is the results of 100, 200, 300, 400, 500 iterations.

iterated through 35 steps, costing 0.9005 seconds of total CPU running time. This shows that our proposed algorithm can get arrive at good results within a short time frame.

## 5.2 Comparisons to ALM in real image

In this section, we compare our proposed method against the ALM solution. The images used in the denoising and inpainting experiments are taken from the Set12 dataset [39], Set14 dataset [40]and the BSD68 dataset [41], and the images in segmentation experiments are from the COVID-CT dataset [42] and PASCAL-VOC2012 dataset [43].

In Fig 6, both the THC algorithm and the proposed algorithm successfully removed noise and avoided the staircase effect. However, the THC model is more difficult to tune due to having additional parameters. As seen in the qualitative results, the dual method maintained edges better.

In order to further compare with the ALM method, we present some quantitative results and convergence times. First, we compare the convergence speed of the two methods in Fig 7. The convergence rate of the proposed method is slightly slower than that of the THC method. The main reason is that the $p^k$ in the calculation of $u^{k+1}$ in 24 uses $u^k$, but the ALM method does not.

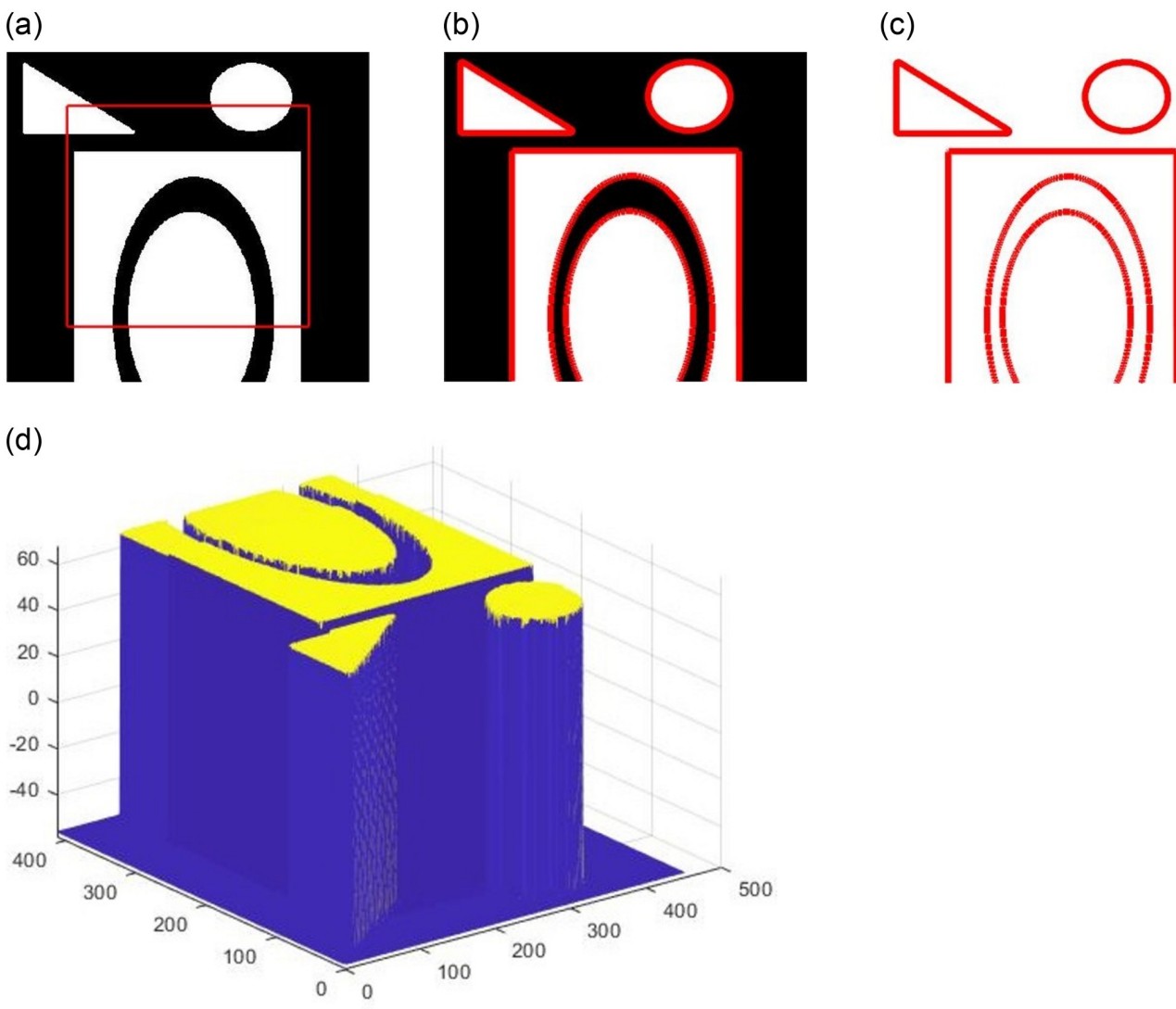

**Fig 3. The segmentation results for synthetic images.** (a) is the original picture with the initialization level set function, (b) is the results of the proposed segmentation method, (c) is the 0 level set function, (d) is the level set function.

In Table 1, we compare the similarity of the denoising results to the ground truths (via PSNR) and the efficiency (via the number of iterations to reach convergence) of the two algorithms over denoising the three images in Fig 6. Four resolutions of each image were used to more thoroughly investigate the two methods. Results show that proposed method requires more iterations to achieve the same quality of denoising, but requires less time each iteration compared to the THC algorithm. For example, for the 256*256 image of 'Baboon', the THC method required 0.081 seconds per iteration whereas the Algorithm 1 only needed 0.013 seconds. The reason why our algorithm requires less time per iteration is the reduction in variables, as our algorithm only uses two variables $u$ and $p$, and neither $u$ nor $p$ needs to be solved iteratively. On the whole, the dual elastica algorithm is far better in efficiency with the same qualitative results.

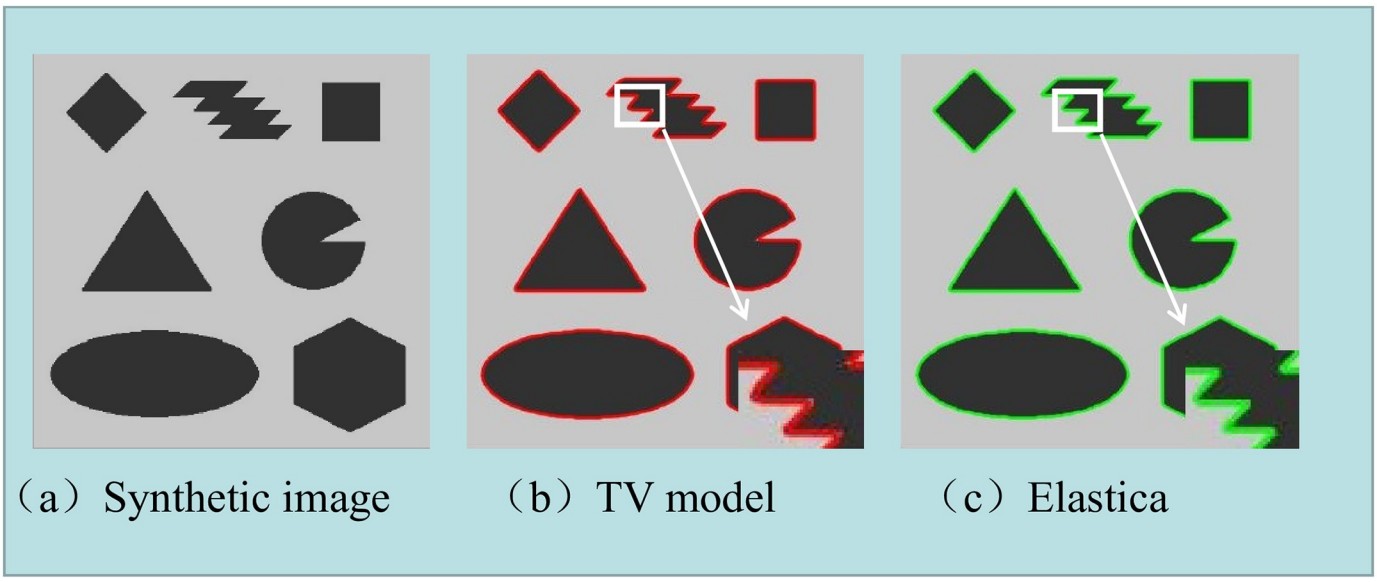

**Fig 4. Image denoising and inpainting results by TV model and Euler's elastica method.**

（a）Synthetic image　　（b）TV model　　（c）Elastica

**Fig 5. Image segmentation results by TV model and Euler's elastica method.**

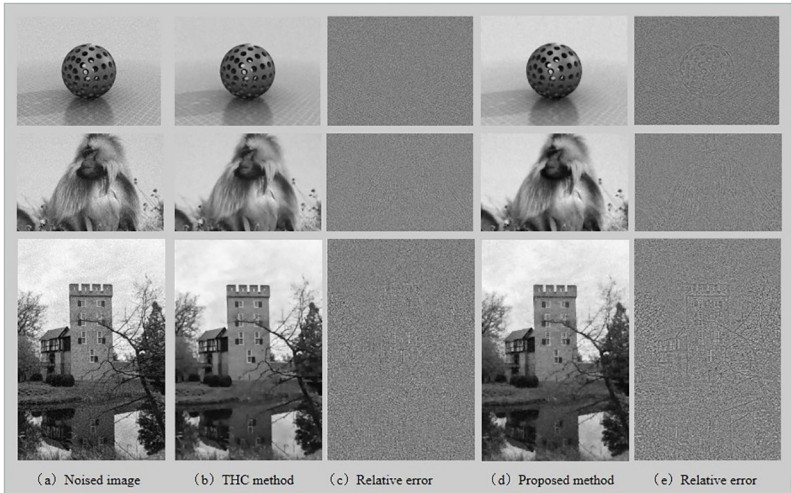

**Fig 6. The denoising results for real images, from top to bottom are 'Ball', 'Baboon', 'Castle'.** (Different from the original picture, it is only for illustrative purposes).

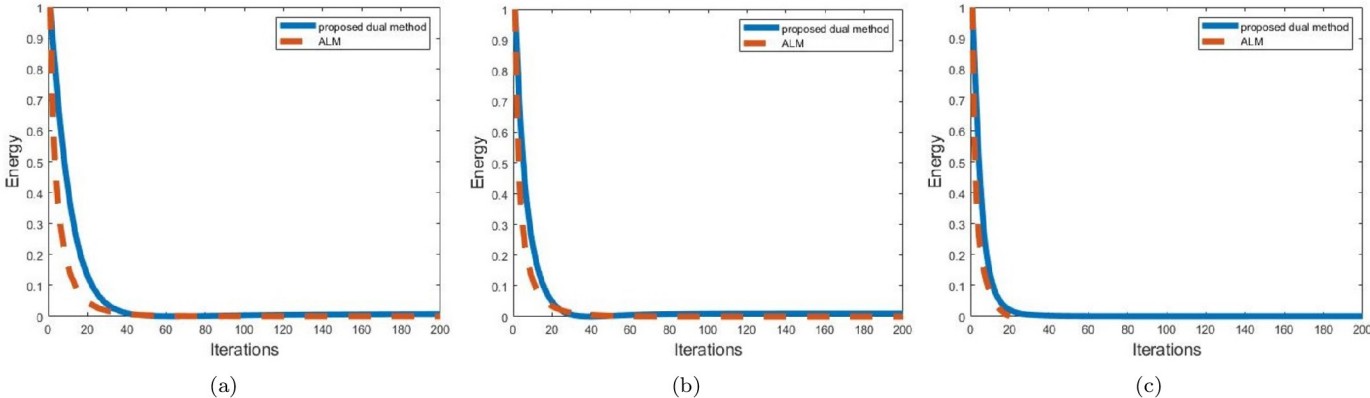

**Fig 7.** The first plot (a) shows the plots of objective function values versus iterations for the example 'Ball'; plot (b) shows the example of 'Baboon'; plot (c) shows the example of 'Castle'.

Fig 8 shows the results of the proposed methods and ZTC method applied to image inpainting, Both methods performed well in repairing damaged areas. However, as can be seen in Table 2, less time is required of our algorithm to produce similar results as the ZTC.

In the next part, we will segment some COVID-CT images to show the effectiveness of our two-phase segmentation algorithm. We used the ZTC algorithm [28] for comparison with our

**Table 1. Performance comparison on different examples and different image sizes using our algorithm and the THC algorithm.**

|  | Ball | | | | Baboon | | | | Castle | | | |
|---|---|---|---|---|---|---|---|---|---|---|---|---|
|  | THC | | Proposed method | | THC | | Proposed method | | THC | | Proposed method | |
|  | PSNR | iters | PSNR | iters | PSNR | iters | PSNR | iters | PSNR | iters | PSNR | iters |
| 64×64 | 23.67 | 6 th | 30.71 | 7th | 28.25 | 7th | 28.37 | 6th | 22.79 | 6th | 29.08 | 5th |
| 128×128 | 25.78 | 5 th | 30.31 | 10th | 28.47 | 6th | 28.25 | 6th | 22.81 | 6th | 29.89 | 6th |
| 256×256 | 28.85 | 3 th | 29.44 | 10th | 29.78 | 6th | 28.95 | 7th | 21.51 | 3th | 28.32 | 6th |
| 512×512 | 29.05 | 3 th | 28.79 | 8th | 30.12 | 5th | 29.24 | 8th | 25.33 | 11th | 29.26 | 9th |

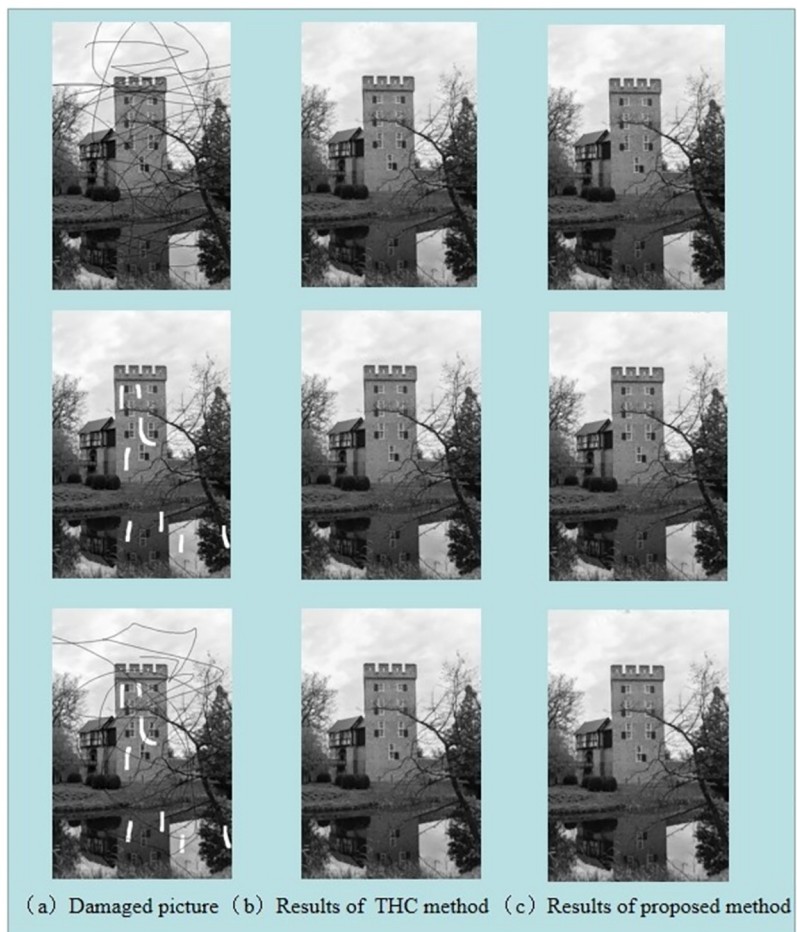

(a) Damaged picture  (b) Results of THC method  (c) Results of proposed method

**Fig 8. The inpainting results for corrupted images of 'Castle'.** (Different from the original picture 'Cameraman', it is only for illustrative purposes).

algorithm. Fig 9 shows two examples of two-phase segmentation of real images. The column (a) is the initialization of the level set function and the original picture, (b) shows the results of the ZTC method, and (c) shows the results of the proposed method. Visually, those results appear similar. However, as shown in Table 3, our algorithm is more efficient in both the execution time per iteration and the total number of iterations.

To further compare the segmentation results numerically, we use the Dice metric to measure the segmentation quality as,

$$DM = \frac{2N_{gs}}{N_g N_s}$$

**Table 2. The number of iterations to reach convergence (iters) and the total CPU time in seconds (CPU(s)) for different images of different sizes by using our algorithm and the THC algorithm in the inpainting problem.**

|  | Cameraman-line | | | | Cameraman-block | | | | Cameraman-mixed | | | |
| --- | --- | --- | --- | --- | --- | --- | --- | --- | --- | --- | --- | --- |
|  | Dual elastica | | THC | | Dual elastica | | THC | | Dual elastica | | THC | |
|  | CPU(s) | iters | CPU(s) | iters | CPU(s) | iters | CPU(s) | iters | CPU(s) | iters | CPU(s) | iters |
| 256×256 | 0.27 | 82th | 0.72 | 32th | 0.75 | 235th | 4.12 | 189 th | 1.25 | 368th | 7.12 | 342th |

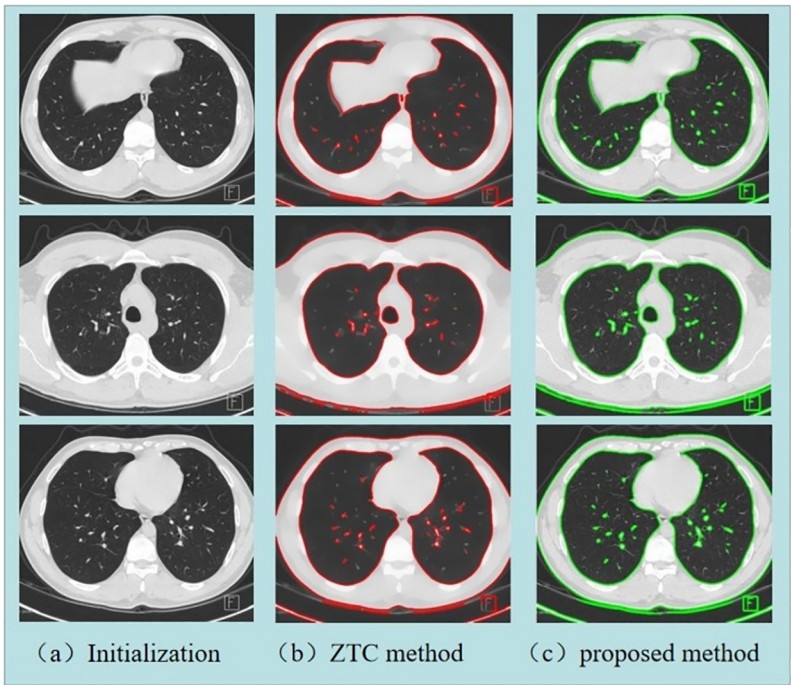

**Fig 9. The segmentation results for CT image.** (Different from the original picture, it is only for illustrative purposes).

**Table 3. The number of iterations to reach convergence (iters) and the total CPU time in seconds (CPU(s)) for different images of different sizes by using our algorithm and the ZTC algorithm in the segmentation problem.**

| | COVID-9 | | | | COVID-82 | | | | COVID-1164 | | | |
|---|---|---|---|---|---|---|---|---|---|---|---|---|
| | ZTC | | Proposed method | | ZTC | | Proposed method | | ZTC | | Proposed method | |
| | CPU(s) | iters | CPU(s) | iters | CPU(s) | iters | CPU(s) | iters | CPU(s) | iters | CPU(s) | iters |
| 256×256 | 4.94 | 4th | 0.86 | 28th | 5.59 | 5th | 31.24 | 29 th | 8.83 | 8th | 22.71 | 21th |
| 512×512 | 5.46 | 5th | 57.99 | 48th | 5.94 | 5th | 24.27 | 20 th | 9.23 | 8th | 26.79 | 22th |

where, $N_{gs}$ is the number of pixels in the object that are correctly segmented, $N_g$ is the number of pixels in the ground truth object, $N_s$ is the number of pixels in the segmented object.

In Table 4, we list the Dice metric numbers obtained in PASCAL-VOC2012 dataset to evaluate the quality of our segmentation results. The data shows that while the proposed method can achieve similar results as the ALM method, the runtime had been reduced to approximately one-sixth.

**Table 4. The number of iterations to reach convergence (iters) and the total CPU time in seconds (CPU(s)) for different images of different sizes by using our algorithm and the ZTC algorithm in the segmentation problem.**

| | Proposed method | | ZTC | |
|---|---|---|---|---|
| | DM | CPU(s) | DM | CPU(s) |
| fighter | 0.9814 | 1.93 | 0.9574 | 7.04 |
| chair | 0.9573 | 1.19 | 0.9485 | 9.32 |
| bottom | 0.9755 | 0.94 | 0.9629 | 6.02 |

## 6 Concluding remarks

In this paper, we used a dual method to solve the Euler's elastica regularizer for image denoising and segmentation. Our method can efficiently reduce the number of parameters, and formulate a more concise algorithm. There are two main contributions. Firstly, by introducing the dual operators, the optimization problem can be divided into simpler sub-problems, and the Chan-Vese model with elastica can be solved more easily. Secondly, our proposed algorithm can effectively reduce the number of parameters, to reduce the dependence on of parameter tuning. Numerical experiments show that compared with the ALM method our algorithm can obtain similar experimental results with less running time.

## Acknowledgments

The author thanks useful comments and valuable suggestions of editors and anonymous commentators.

## Author Contributions

**Data curation:** Jintao Song, Huizhu Pan, Jieyu Ding, Zhenkuan Pan.

**Formal analysis:** Jintao Song, Zhenkuan Pan.

**Methodology:** Jintao Song, Zhenkuan Pan.

**Resources:** Jintao Song, Huizhu Pan, Weibo Wei, Zhenkuan Pan.

**Supervision:** Jieyu Ding, Weibo Wei, Zhenkuan Pan.

**Validation:** Jintao Song.

**Writing – original draft:** Jintao Song, Huizhu Pan.

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
