## [Decision Letter · Decision Letter 0]

7 Oct 2021

PONE-D-21-19897

New Dual Method for Elastica Regularization

PLOS ONE

Dear Dr. Pan,

Thank you for submitting your manuscript to PLOS ONE. After careful consideration, we feel that it has merit but does not fully meet PLOS ONE’s publication criteria as it currently stands. Therefore, we invite you to submit a revised version of the manuscript that addresses the points raised during the review process.

Please pay careful attention to the journal and Editor comments below the signature.

We look forward to receiving your revised manuscript.

Kind regards,

Paul Atzberger 

Academic Editor

PLOS ONE

Journal Requirements:

We note that in this manuscript you have presented an image known as Lena/Lenna, which has a problematic history, please see https://en.wikipedia.org/wiki/Lenna for more information. We do not feel that this image is in line with the values of PLOS ONE, and would therefore request that you at this point substitute this image in the manuscript with another image.

Additional Editor Comments (if provided):

Overall, the paper appears to be well written, but there were a few minor typos throughout, such as "relevent error" which should probably be "relative error." Please be sure to do a final careful proof-reading of for the final version of the manuscript.

Reviewers' comments:

Reviewer's Responses to Questions

**Comments to the Author**

1. Is the manuscript technically sound, and do the data support the conclusions?

Reviewer #1: Yes

2. Has the statistical analysis been performed appropriately and rigorously? 

Reviewer #1: Yes

3. Have the authors made all data underlying the findings in their manuscript fully available?

Reviewer #1: Yes

4. Is the manuscript presented in an intelligible fashion and written in standard English?

Reviewer #1: Yes

5. Review Comments to the Author

Reviewer #1: The paper is well written and adds to existing literature on the subject. The introduction is well composed and has been developed on the right lines. Most appropriate methodology has been used for analysis of data and information. The results have been reported well. Logical sequence of interpretation has been followed and developed scientifically. The discussion has been well brought out and the cogency of arguments are well thought out. The discussion is comprehensive and complete. References are as required. May be accepted for publication.

6. PLOS authors have the option to publish the peer review history of their article (what does this mean?). If published, this will include your full peer review and any attached files.

Reviewer #1: No

---

## [Author Response · Author response to Decision Letter 0]

30 Oct 2021

Many thanks to the editor and the reviewer for affirming our work and pointing out the potential issues behind the Lena image. We apologize for our misuse of the image and have removed it from our paper to comply with the standards of the PLOS ONE journal. We have also carefully revised our manuscript according to the comments.

Thank you again for your feedback and consideration.

Best Regards

---

## [Editor Report · Decision Letter 1]

25 Nov 2021

New Dual Method for Elastica Regularization

PONE-D-21-19897R1

Dear Dr. Pan,

We’re pleased to inform you that your manuscript has been judged scientifically suitable for publication and will be formally accepted for publication once it meets all outstanding technical requirements.

Kind regards,

Paul J Atzberger, Ph.D.

Academic Editor

PLOS ONE
---

## [Editor Report · Acceptance letter]

10 Jan 2022

PONE-D-21-19897R1 

New Dual Method for Elastica Regularization 

Dear Dr. Pan:

I'm pleased to inform you that your manuscript has been deemed suitable for publication in PLOS ONE. Congratulations! Your manuscript is now with our production department. 

Kind regards, 

on behalf of

Dr Paul J Atzberger 

Academic Editor

PLOS ONE